# Reduced myeloid commitment and increased uptake by macrophages of stem cell–derived HPS2 neutrophils

Steven DS Webbers[1,3] , Cathelijn EM Aarts[1], Bart Klein[1], Dané Koops[1,3] , Judy Geissler[1] , Anton TJ Tool[1], Robin van Bruggen[1], Emile van den Akker[2] , Taco W Kuijpers[1,3]

Hermansky–Pudlak syndrome type 2 (HPS2) is a rare autosomal recessive disorder, caused by mutations in the *AP3B1* gene, encoding the $\beta$3A subunit of the adapter protein complex 3. This results in mis-sorting of proteins within the cell. A clinical feature of HPS2 is severe neutropenia. Current HPS2 animal models do not recapitulate the human disease. Hence, we used induced pluripotent stem cells (iPSCs) of an HPS2 patient to study granulopoiesis. Development into CD15[POS] cells was reduced, but HPS2-derived CD15[POS] cells differentiated into segmented CD11b[+]CD16[hi] neutrophils. These HPS2 neutrophils phenocopied their circulating counterparts showing increased CD63 expression, impaired degranulation capacity, and intact NADPH oxidase activity. Most noticeable was the decrease in neutrophil yield during the final days of HPS2 iPSC cultures. Although neutrophil viability was normal, CD15[NEG] macrophages were readily phagocytosing neutrophils, contributing to the limited neutrophil output in HPS2. In this iPSC model, HPS2 neutrophil development is affected by a slower rate of development and by macrophage-mediated clearance during neutrophil maturation.

## Introduction

Hermansky–Pudlak syndrome (HPS) is an extremely rare autosomal recessive disorder, affecting ~1 in 1,000,000 people worldwide. It was characterized in patients with ocular albinism (hypopigmentation), prolonged bleeding because of a platelet abnormality, and pigmented macrophages in the bone marrow ([1], [2]). The disorder is genetically heterogeneous and caused by mutations in genes that mostly function in membrane and protein trafficking ([3]). In humans, there are 10 genes known that can lead to a particular HPS phenotype, where the different subtypes have their own distinguishing clinical features ([3], [4]). For HPS type 2 (HPS2), this includes immunodeficiency, where congenital neutropenia is a distinguishing characteristic ([5], [6], [7], [8]), resulting in an increased susceptibility to infections.

HPS2 is caused by mutations in the *AP3B1* gene, encoding the $\beta$3A subunit of the adapter protein complex 3 (AP-3), affecting the expression and functionality of AP-3 ([9]). The AP-3 facilitates trafficking of vesicles from the *trans*-Golgi network and/or endosomal compartments to endosome/lysosome-related organelles ([10], [11], [12]). Cells deficient in AP-3 show disturbed formation of lysosomes and storage of some of the granular components.

Currently, there are a few animal models described to study HPS2 in more depth, for example, the *Pearl* strain mouse model and canine models, but these models do not phenocopy the neutrophil defect in patients ([13], [14], [15]). To study the lack of proper neutrophil development in HPS2 for humans, we improved our previously published human iPSC-derived neutrophil model ([16]), and used this to generate and validate the phenotype of HPS2 iPSC-derived neutrophils from a previously published HPS2 induced pluripotent stem cell (iPSC) ([17]) line derived from HPS2 patient material compared with a healthy control iPSC line ([17]). Here, we demonstrate perturbed commitment toward CD15[+] myeloid precursors of the HPS2 iPSC hematopoietic progenitors compared with WT, but once committed, HPS2 iPSC-derived CD34[+] hematopoietic myeloid progenitors do show normal neutrophil development and full segmentation of nuclei with unperturbed viability when compared to WT iPSC-derived neutrophils. Key features for circulating HPS2 neutrophils, such as an altered granule compartment and high basal CD63 expression compared, were also found on the HPS2 iPSC-derived neutrophils.

In addition, HPS2 patients can be hospitalized with, in some cases lethal, hemophagocytic lymphohistiocytosis in the absence of obvious triggers ([18], [19]). Although HPS2 iPSC neutrophils phenocopied circulating HPS2 neutrophils with an altered granule compartment ([6]), the HPS2 iPSC cultures show limited outgrowth of macrophages and unexpected uptake of CD15[+] myeloid cells, and, taken together with the reduced myeloid commitment, could well explain the observed neutropenia in HPS2.

[1]Department of Molecular Hematology, Sanquin Research, Amsterdam University Medical Center (AUMC), University of Amsterdam, Amsterdam, Netherlands [2]Department of Hematopoiesis, Sanquin Research Amsterdam, Amsterdam, Netherlands [3]Department of Pediatric Immunology, Rheumatology & Infectious Diseases, Emma Children's Hospital, AUMC, University of Amsterdam, Amsterdam, Netherlands

Correspondence: S.webbers@sanquin.nl; t.w.kuijpers@amsterdamumc.nl

# Results

We previously generated iPSCs from an HPS2 patient and unrelated healthy donors as described before (17). The HPS2 patient–derived and control-derived iPSC lines showed pluripotency, whereas the HPS2 iPSC line still maintained the mutation in the *AP3B1* gene as the original BOECs derived from the patient (17). In contrast to the WT iPSC line, the HPS2 iPSC line showed a complete lack of the AP3β1 subunit protein (Fig 1A), and as a consequence also other proteins of AP-3 such as the AP3μ subunit (6, 17, 20, 21).

### HPS2-derived neutrophils: limited expansion capacity

To differentiate iPSCs toward neutrophils, we used a feeder-free, single iPSC-derived monolayer differentiation system that is divided into two main phases, colony differentiation and hematopoietic specification, followed by the subculture phase in which the hematopoietic cells are harvested and subsequently matured to generate fully segmented neutrophils (Fig S1) (16). In the subculture phase (i.e., terminal maturation), we observed that the cell expansion per differentiated colony was lower in the HPS2 iPSCs compared with the WT lines at the final day of differentiation at day 6 (Fig 1B), with a surprisingly rapid decline starting at day 3 in the subculture phase (Fig 1C). This decline in expansion was continued when extending the subculture phase to day 8. Cell death by classical apoptosis, which was determined by staining for both Annexin V and Hoechst staining (gating strategy in Fig S2A–C), was not involved, because HPS2 viability was comparable with WT (Fig 1D), which is in line with the inability of the pan-caspase inhibitor z-VAD to rescue this early demise (Fig S3A).

Moreover, we looked at myeloid commitment by monitoring the switch from CD34 expression, a hallmark of early hematopoietic progenitors, to CD15 expression, a hallmark of early granulocyte progenitors (Fig 1E and F) (22, 23). Notably, HPS2-90 cultures still showed an increased number of cells expressing CD34 at day 3 of the subculture and an overall lower expression of CD15 during the entire subculture, which could not be explained by an increased number of CD15$^{NEG}$ cell monocytes instead. Both WT and HPS2 cultures showed similar CD14 geometric mean fluorescence intensities (gMFI) on CD15$^{NEG}$ cells as blood-derived PMNs, as well as comparable histograms (Fig 1F and G). Taken together, this suggests an early block in commitment to the myeloid compartment.

### HPS2 iPSC-derived mature neutrophils show fully differentiated morphology and phenotype

Despite the early block in neutrophil commitment, both HPS2 and WT iPSC lines included cells with the characteristic segmented nucleus of mature neutrophils (Fig 2A). Interestingly, CD11b/CD16 expression on CD15$^{POS}$ cells was similar between both HPS2 and WT cultures (Fig 2B and C), indicative of a fully mature neutrophil phenotype. For neutrophils, the expression of SIGLEC9 and EMR3 is used as maturity markers (23, 24, 25, 26) (Fig S4A–E). HPS2 iPSCs showed an EMR3 expression pattern similar to WT iPSCs (Fig 2D). When focusing on the SIGLEC family of proteins with a diversified expression among leukocytes, we assessed the levels of SIGLEC9

(for neutrophils, Fig 2D) versus SIGLEC8 (for eosinophils, Fig S5A). Although the percentage of SIGLEC9$^{POS}$ cells was comparable between HPS2 and WT neutrophils, expression levels on HPS2 iPSC neutrophils were much higher than WT (Fig 2D, P < 0.01). Although SIGLEC9 crosslinking has been found to induce apoptosis (27), viability was not changed (Fig 1D) at any stage of subculture, and the addition of a pan-caspase inhibitor compared with solvent did not show any effect on WT-30– or HPS2-derived cell yield, survival, or maturation (Fig S3A and B).

Within the CD15$^{POS}$CD11b$^{POS}$CD16$^{NEG}$ cell fractions, the HPS2 iPSC-derived cultures showed a slightly increased expression of SIGLEC8 for HPS2 cells compared with WT (Fig S5A), despite the absence of eosinophil-inducing factors like IL-3 and GM-CSF in the final phase of cell culture (28, 29). Histograms show a small part of the HPS2 culture having similar SIGLEC8 levels compared with blood-derived eosinophils. Our iPSC-derived neutrophils showed an almost complete absence of CD123 and CD163 (Fig S5B), and low CD14 expression compared with healthy donor M-CSF macrophages (Fig S5C).

Because of the presence of fully segmented cells on cytospins and the normal surface marker expression of most myeloid markers, we conclude that HPS2 iPSC neutrophils did reach end-stage maturation. These findings argue against a maturation defect for HPS2 iPSC neutrophils per se.

### HPS2 iPSC-derived neutrophils show a granule defect similar to circulating HPS2 neutrophils

We also evaluated neutrophil effector functions for the HPS2 and WT iPSC neutrophils. We observed that HPS2 iPSC-derived cultures showed lower ROS production when measured unsorted upon activation by different stimuli (Fig 2E). Although the unsorted HPS2 iPSC-derived cells do contain proteases similar to WT iPSC-derived cells, they were unable to release them (Fig 2F). This corresponds to the lack of CD63 up-regulation from azurophilic granules or CD66b from specific granules of mature neutrophils (Fig 2G) (30, 31, 32). Moreover, CD63 was already strongly increased on HPS2 iPSC-derived neutrophils. Taken together with the degranulation defect, these defects are highly reminiscent of previous findings with circulating HPS2 neutrophils related to the defective lysosomal sorting by the AP-3 defect, as we have shown previously (6, 33).

### HPS2 iPSC-derived myeloid cells show macrophage outgrowth

During the subculture phase, we noticed that HPS2 iPSC cultures showed large cell aggregates, which were absent in the WT counterpart (Fig 3A). In addition, the most remarkable on the cytospins from HPS2 iPSCs was the number of large macrophages, which were almost completely absent on the cytospins from the WT iPSCs (Fig 3B). This clearly contrasted with the WT cultures. HPS2 macrophages were characterized by their very large size, round nuclear morphology, lipid droplets, and uptake of other cells. Particularly, this last feature is reminiscent of bone marrow smears from HPS2 patients with hemophagocytic episodes (19). Outgrowth of macrophages from iPSCs is possible in the presence of IL-3 and FSC, as also used in our culture system (Fig S1) (34).

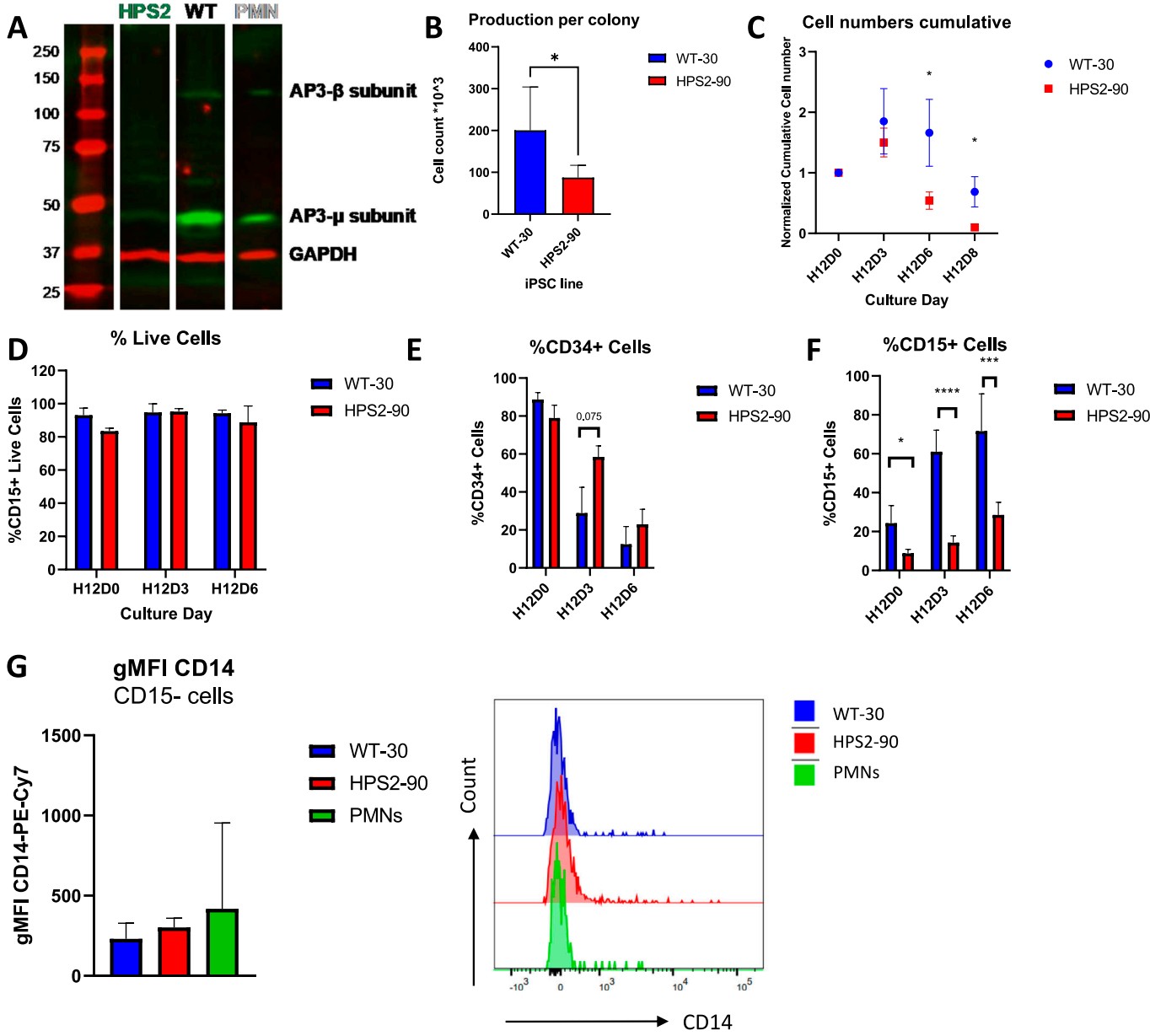

**Figure 1. Comparison of expansion and viability of HPS2 and WT iPSC-derived neutrophil-like cells.**
**(A)** Western blot detection of AP3?1 (140 kD) and AP3μ1 (47 kD) subunit proteins, both indicated in green, in whole iPS cell lysates from the HPS2 and WT iPSC lines. Whole-cell lysates of neutrophils derived from blood (PMN) were taken along as a control. Glyceraldehyde-3-phosphate dehydrogenase (GAPDH) was used as a loading control. **(B)** Cell count at H12D6 divided by the total number of colonies seeded (n = 6-7). HPS2-90 cultures show lower cell counts per colony at day 28 compared with WT iPSC. Data are shown as the mean + SD. Statistical differences were determined by a paired $t$ test (*$P$ < 0.05). **(C)** Cumulative cell counts, normalized to cell counts on the first day of subculture (n = 6–7). HPS2-90 cultures show a marked decrease in cell count at H12D6 and H12D8 compared with WT. Data are shown as the mean + SD. Statistical differences were determined with a multiple, corrected, paired $t$ test (*$P$ < 0.05). **(D)** Percentage of live CD15[POS] cells for each day of subculture (n = 6–7). Cells were determined to be viable based on the absence of both PS exposure measured with Annexin V staining and an increase in nuclear DNA staining with Hoechst 33343 by FACS measurement. Data are shown as the mean + SD. Statistical differences were determined with a multiple paired $t$ test. **(E)** Percentage of CD34[POS] cells, decreasing over 6 d of subculture (n = 6–7). HPS2-90 iPSCs switch less rapidly to CD34[NEG] compared with WT. Data are shown as the mean + SD. Statistical differences were determined with a multiple, corrected, paired $t$ test (*$P$ < 0.05, ***$P$ < 0.001, and ****$P$ < 0.0001). **(F)** Percentage of CD15[+] cells, increasing over the 6 d of subculture (n = 6–7). Overall switch to CD15[POS] is lower in HPS2-90 cultures compared with WT. Data are shown as the mean + SD. Statistical differences were determined with a multiple, corrected, paired $t$ test. (*$P$ < 0.05, ***$P$ < 0.001, and ****$P$ < 0.0001). **(G)** GMFI of CD14 expression on CD15[NEG] cells of WT-30, HPS2-90, and circulating PMNs, accommodated with representative histograms for each cell type (n = 6–7). Data are shown as the mean + SD.
Source data are available for this figure.

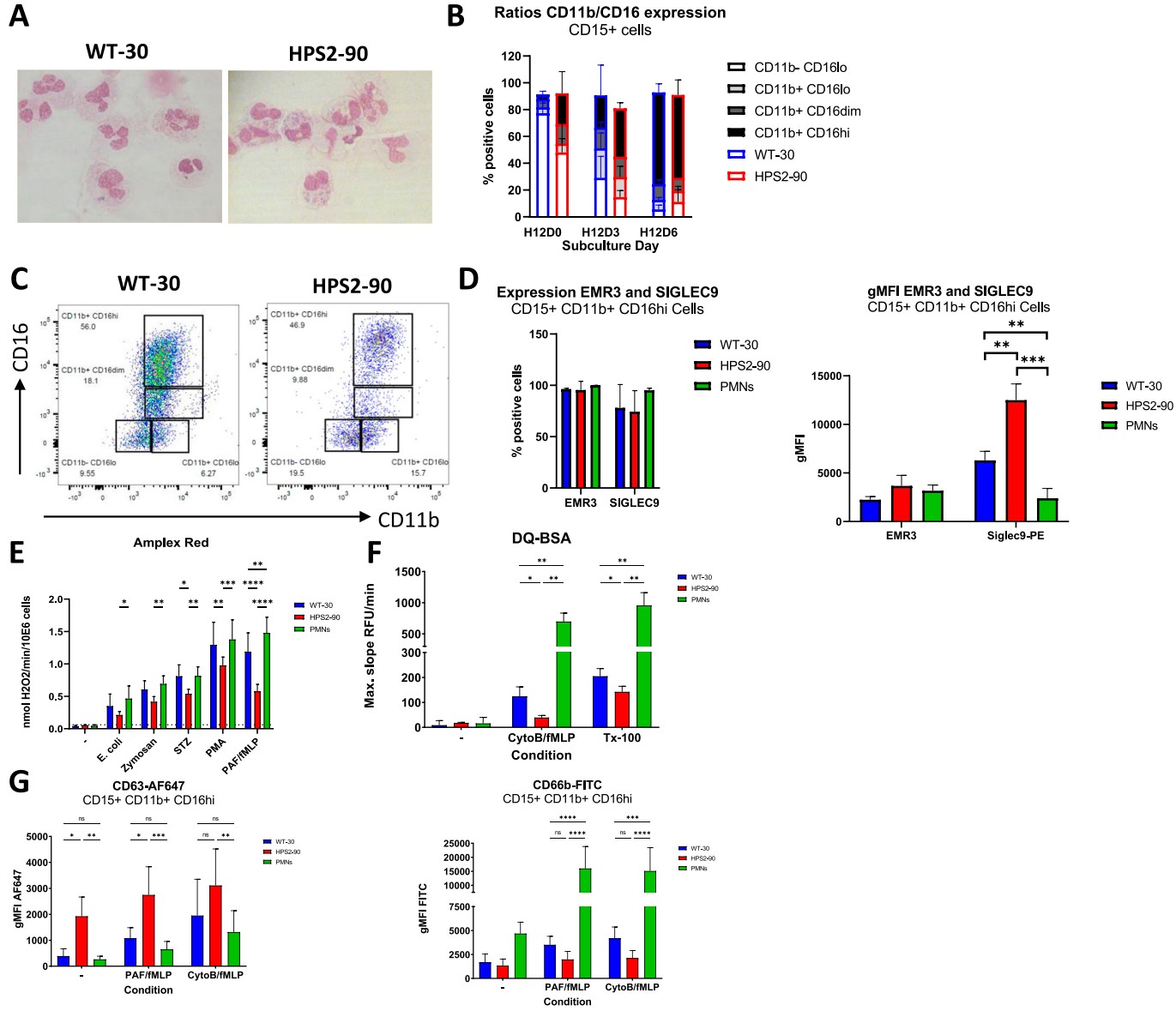

**Figure 2. Morphology, phenotype, and functionality of iPSC-derived neutrophils.**
**(A)** Representative image of cytospins with either WT-30 or HPS2-90 cells showing fully segmented nuclei. Scale bar set at 10 μm. **(B)** Ratio of CD11b to CD16 expression over the subculture phase (n = 6–7). Data are shown as the mean + SD. **(C)** Representative plots for CD11b and CD16 on the sixth day of subculture. A clear increase in CD11b and CD16 indicates neutrophil maturation (n = 4). **(D)** Percentages of neutrophil markers EMR3 and SIGLEC9 on CD15[POS]CD11b[POS] and CD16[hi] cells (n = 4). Data are shown as the mean + SD. Statistical differences were determined using a multiple comparison test (**$P < 0.01$ and ***$P < 0.001$). **(E)** ROS production measured as the conversion of Amplex Red to fluorescent resorufin by $H_2O_2$. Depicted is the maximal slope of 2 min measured within 30 min (n = 6–7). Data are shown as the mean + SD. Statistical differences were determined using a multiple comparison test (*$P < 0.05$, **$P < 0.01$, and ****$P < 0.0001$). **(F)** Protease release (elastase and cathepsin-G) defined by cleaving of DQ from DQ-BSA. The maximal slope during 30 min of measurement is depicted here (n = 4). Data are shown as the mean + SD. Statistical differences were determined using a multiple comparison test (*$P < 0.05$ and **$P < 0.01$). **(G)** After 10-min stimulation, the surface expression of CD63 (azurophilic granule) or CD66b (specific granule) is measured by FACS analysis on CD15[POS]CD11b[POS]CD16[hi] cells (n = 6). Data are shown as the mean + SD. Statistical differences were determined using a multiple comparison test (*$P < 0.05$, **$P < 0.01$, and ***$P < 0.001$).

Within these HPS2 macrophages, we observed the presence of smaller cells and nuclear cell debris, suggesting cell uptake during the culture (Fig 3B). As the viability of the HPS2 cells in our cultures was comparable to the WT, the uptake of cells by macrophages in the cell culture might explain the overall low yield observed at the end of differentiation of the HPS2 iPSC cultures (Fig 1B). This phenomenon was also observed when the iPSCs were cultured in

the presence of the pan-caspase inhibitor z-VAD (Fig S3C), again demonstrating that the "eat-me" signal seemed to be unrelated to apoptosis as such.

Because of the sharp decline observed in the HPS2 cell count at the final stage of the subculture phase, and the concomitant large phagocytosing macrophages on our cytospins, we opted to perform a co-culture of M-CSF monocyte-derived macrophages together

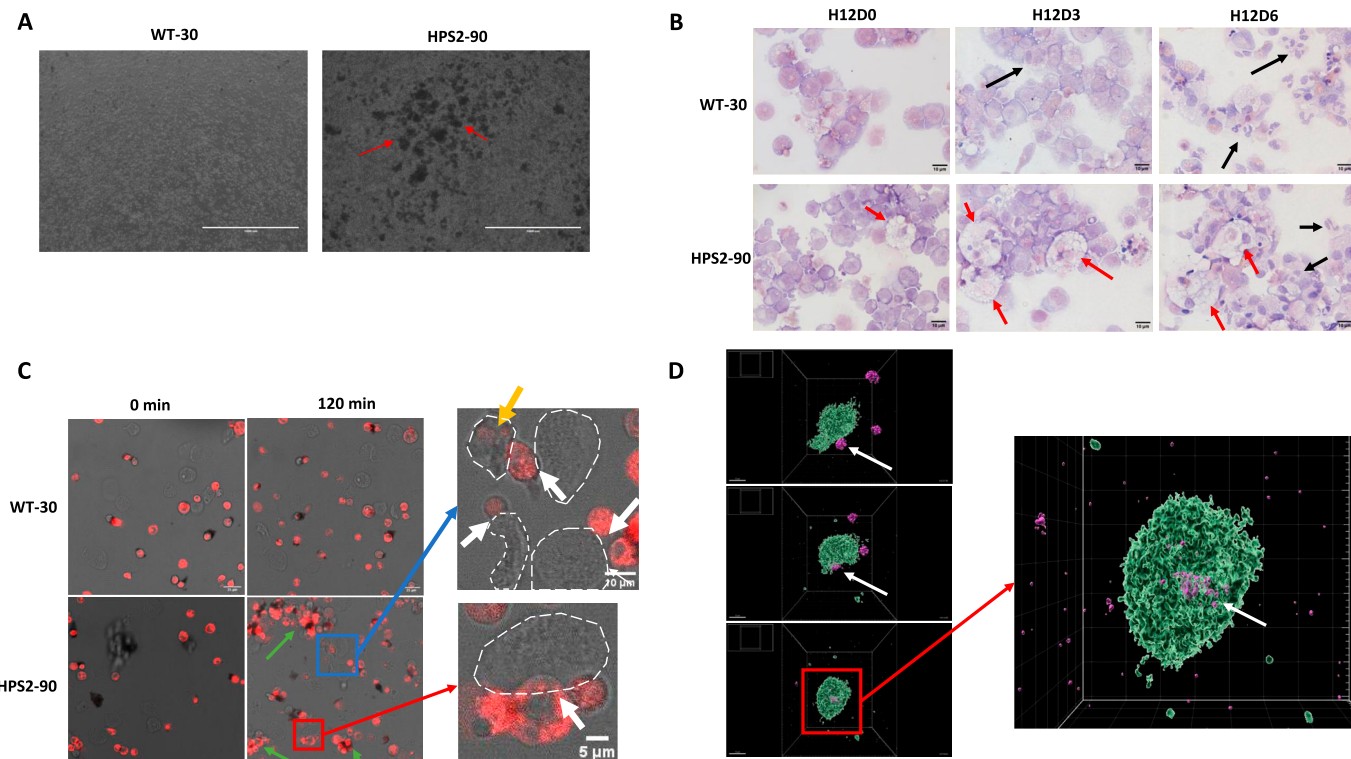

**Figure 3. Uptake of HPS2 cells in culture and by M-CSF-derived macrophages from healthy donors.**
**(A)** Images taken on day 3 of the subculture phase. HPS2 cultures contain large clusters of cells (red arrows). Scale bar set at 1,000 $\mu$m. **(B)** May–Giemsa–stained slides of iPSC-derived hematopoietic progenitors (day 0) differentiating toward neutrophils (day 6). Scale bar set at 10 $\mu$m. Upon 6 d of G-CSF–driven differentiation, segmentation of nuclei increases (black arrows). HPS2 iPSC slides show large macrophages from early on, which increase in number over 6 d (red arrows). **(C, D)** WT-30 or HPS2-90 cells from day 3 of the subculture were added to blood monocyte–derived M-CSF macrophages, and images were taken every minute for 2 h. **(C)** After 2 h, large cell clusters were visible (green arrows) in HPS2-90 co-cultures, but not WT-30. Interactions (white arrows) between HPS2-90 cells and M-CSF macrophages (encircled, white dashed lines) can be readily spotted, being almost absent in WT co-cultures. The orange arrow shows internal red fluorescent signals of two cells inside of an M-CSF macrophage, indicating phagocytosis of HPS2-90 cells. These events were absent in WT co-cultures. Scale bars set at 25, 10, and 5 $\mu$m. **(D)** Confocal images of an HPS2 cell being taken up by an M-CSF macrophage. Images show the progress of the HPS2 cell interacting and subsequently being internalized by the M-CSF macrophage. Scale bar set at 15 $\mu$m.

with either HPS2 iPSC-derived cells or WT cells on day 3 of sub-culture. Both iPSC lines were stained with CellTracker Red and co-cultured with the M-CSF macrophages for 2 h and monitored by live-cell imaging (Fig 3C, Video 1 and Video 2). HPS2 iPSC-derived cells were forming large clusters of cells within 2 h and also interacted with the M-CSF macrophages, which was much less evident with WT cells. In agreement with the data in Fig 3B and C and in contrast to WT cells, confocal imaging showed that M-CSF macrophages internalized HPS2-derived cells (Fig 3D, Video 3 and Video 4). These data suggest an active process in which the HPS2 cells are being engulfed by macrophages, either from healthy donor macrophages or by macrophages that develop in the HPS2 iPSC culture and clear away a large part of the differentiating neutrophils.

## Discussion

In this study, we used human HPS2 iPSCs instead of the non-informative animal models, as a model system to study neutrophil development in this very rare congenital neutropenia disorder (3, 5, 13, 14, 15) The neutropenia in HPS2 seems to be a consequence of

both a reduced rate of myeloid differentiation and macrophage-mediated phagocytosis.

We observed that HPS2-derived stem cells were not able to commit to the myeloid–neutrophil cell lineage at similar rates as WT iPSCs, but still have the full potential to differentiate into mature neutrophils based on morphology, phenotype, and to a large extent their functionality. This potential clearly differs from the differentiation arrest at the promyelocyte stage in severe neutropenia by *ELANE* mutations (35).

The larger CD15$^{NEG}$ cell population in HPS2 iPSC cultures seems not to be due to skewing toward CD14$^+$ cells but suggests delayed differentiation of the CD15$^{NEG}$ population into CD11b$^{POS}$ meta-myelocytes and neutrophils. A reduction in neutrophil elastase (NE) content has been reported in HPS2 (5, 33). NE is normally stored in azurophilic granules and has been suggested, by its misrouting in AP-3–deficient cells, to cause the chronic neutropenia observed in these patients (5, 36). We did not observe a decrease in total NE content in iPSC-derived HPS2 neutrophils (6), but iPSC HPS2 cells were unable to release NE (Fig 2F). Although the expression of CD63 on iPSC-derived neutrophils was increased, similar to circulating patient neutrophils, the elevated SIGLEC9 was not (6).

The HPS2 cell population after differentiation also included macrophages, which clearly showed the presence of phagocytosed cells and cell remnants (Fig 3B). Studies into iPSC macrophage differentiation have shown that macrophage differentiation from HSPCs occurs already in the presence of IL-3 and FCS (28, 34), which is also present in our setup. As these components are also essential for outgrowth of neutrophil progenitors, some macrophage contamination is to be expected. In addition, because we found that the CD15[+] myeloid commitment was blocked compared with WT, it is possible that part of the HPS2 culture differentiated into macrophages instead.

The massive uptake of cells and cell remnants by macrophages in the final 3 d of the iPSC subcultures of neutrophils can to a large extent explain the overall low yield observed at the end of differentiation of the HPS2 cells compared with WT. The expression of proapoptotic SIGLEC9 (27) may result in early clearance by macrophages (18). However, we did not observe early neutrophil apoptosis or a lack of cell viability for the HPS2 iPSCs throughout the subculture based on flow cytometry compared with WT iPSCs (Fig 1D). Moreover, the addition of the pan-caspase inhibitor z-VAD did not save the relative cell counts of the subculture (Fig S3A), ruling out caspase-mediated cell death as the cause. In addition, increased SIGLEC9 on circulating neutrophils can result in both a caspase-dependent and a caspase-independent cell death, the second that is characterized by high DNA staining but low Annexin V staining (27). Such a population was also absent from our cultures, as over 80–95% of all cells during every measured timepoint of subculture stained negative for Annexin V and had low nuclear DNA staining by Hoechst (Fig 1D). Instead, the massive uptake by macrophages in the HPS2 iPSC cultures of cells, presumed to be myeloid of nature, was observed from day 3 onward, as shown by our widefield and confocal imaging. Whether neutrophil SIGLEC9 crosslinking in contact with macrophages may have facilitated their uptake can only be speculated upon.

In clinical terms, strong activation of macrophages can result in a clinical syndrome of hemophagocytosis (37). Under conditions of inflammation, such hemophagocytic reactions may culminate in severe and often fatal disease manifestations with the active release of proinflammatory cytokines. Clinical hemophagocytosis has also been described in HPS2 patients (18). We believe, however, that based on our data presented here, this is not reminiscent of activated macrophages in a process of hemophagocytosis. Instead, the data suggest that HPS2 neutrophils carry signals for uptake as demonstrated with control M-CSF macrophages. Which factor(s) of the many efferocytosis signals is(are) decisive in this process remains unclear to date.

To conclude, in our HPS2 iPSC model we observed HPS2 characteristics that were in line with studies investigating mature neutrophils derived from the blood of HPS2 patients (6, 33). HPS2 iPSC-derived cells showed a lower rate of full commitment toward the neutrophil cell lineage, although overall viability was not affected, and an inherent tendency to eliminate neutrophils during their maturation. The HPS2 iPSC cultures are thus reminiscent of some of the clinical characteristics of the severe chronic neutropenia in patients.

# Materials and Methods

## iPSC line generation and cell isolation

The iPSC lines from the HPS2 patient (SANi009-A; compound heterozygous c.177delA and c.1839-1842delTAGA) and healthy donor (SANi010-A) were cultured as previously described (17). For these studies, patient cells were generated and biobanked with the informed consent of the patient, and parents, informed and signed according to our hospital's regulation (Amsterdam UMC, iPSC Biobank, version 5, April 2022).

For the isolation of mature neutrophils from blood, heparinized blood was obtained, after informed consent and according to the Declaration of Helsinki, from healthy volunteers. Blood leukocytes were separated based on density by centrifugation over isotonic Percoll (Pharmacia) with a specific density of 1.076 g/ml. Neutrophils were obtained from the pellet fraction after two rounds of erythrocyte lysis with hypotonic ammonium chloride solution (4.15 g $NH_4Cl$, Merck; 0.5 g $KHCO_3$, Merck; 18.5 mg EDTA; Merck; and 500 ml WFI; Gibco) at 4°C for 5–10 min, essentially as described previously (38).

For the isolation of monocytes to culture into macrophages, after density separation by Percoll, PBMCs were taken from the ring fraction. PBMCs were washed with PBS, and monocytes were isolated by MACS separation with CD14 microbeads (Miltenyi Biotec). Monocytes were differentiated toward macrophages on Ibidi slides in IMDM (Gibco) with 10% FCS and 50 ng/µl M-CSF (PeproTech) for 10 d. Half-medium changes were performed every 3–4 d.

## iPSC colony differentiation toward neutrophils

The iPSC lines SANi009-A (HPS2-90) and SANi010-A (WT-30) were maintained feeder-free on Matrigel (Corning)-coated plates in E8 medium (Gibco), as reported previously (16). For differentiation, iPSCs were single-cell–seeded at a low density of 250 cells for SANi010-A and 1,200 cells for SANi009-A on Matrigel-coated dishes in E8 medium supplemented for the first 4 d with RevitaCell (Thermo Fisher Scientific). The differentiation was performed as described previously (16), with some adaptations listed here and visualized in Fig S1. After 10 d of single colony formation, dishes were scraped mechanically to allow 10–20 colonies per plate with ample space for outgrowth. From day 0 to 6, cultures received Stemline II (Sigma-Aldrich) supplemented with 1% Pen/Strep, 10 ng/ml bFGF, 20 ng/ml of BMP4, and 40 ng/ml of VEGF. From day 6 of differentiation, cultures received Stemline II with 1% Pen/strep, 1 ng/ml IL-3, 10 ng/ml IL-6, 10 ng/ml VEGF, 20 ng/ml BMP4, 50 ng/ml FLT-3, 50 ng/ml hSCF (manufactured in-house), and ITS (1:100). From day 12, suspension cells were cultured in Stemline II with 1% Pen/strep, 30 ng/ml G-CSF (Neupogen, clinical grade), 50 ng/ml FLT-3, 50 ng/ml hSCF, and 10% heat-inactivated serum (HyClone Fetal-Clone I Serum; Thermo Fisher Scientific). From day 15, cells were cultured in Stemline II with 1% Pen/strep, 30 ng/ml G-CSF, and 10% heat-inactivated serum. All growth factors were purchased from STEMCELL Technologies unless indicated otherwise. In case z-VAD (BD Pharmingen) was used, a final concentration of 20 µM was

added on days 12 and 15 of differentiation, with a final DMSO concentration of 0.1%.

## Antibodies and flow cytometry

Antibodies are listed in Table S1. Flow cytometry data were measured using a Canto II flow cytometer (BD Biosciences) and analyzed using FlowJo software (version 10.7 and 10.8; Tree Star). For neutrophils and cultured neutrophils, an example of the gating strategy is presented in Fig S2. Attempts at flow cytometric analysis of the iPSC macrophages proved particularly challenging because of aggregation and high fluorescent background (data not shown).

## May–Giemsa staining

50,000 cells were spun at 1,000 RPM for 10 min (Shandon CytoSpin 4 Cytocentrifuge, Shandon cytospin sealed head rotor [Order No. 59910018]) onto 76 × 26 mm glass microscope slides. The slides were air-dried and subsequently stained for 5 min in May–Grünwald (Merck), 15 min in phosphate buffer, and 30 min in Giemsa (Merck) solution. Slides were rinsed between steps with deionized water and finally air-dried.

## Immunoblot analysis

For lysates, 20 $\mu$l contained 1.0 × 10$^6$ cells, regardless of the cell type. Samples were separated on 7.5% or 10% sodium dodecyl sulfate–polyacrylamide gel electrophoresis (SDS–PAGE) made in-house and subsequently transferred to nitrocellulose membranes by wet transfer (Schleicher & Schuell). Membranes were blocked with 5% non-fat milk in TBS-T (0.1%) (Elk Campina). The following antibodies were used for detection: monoclonal rabbit anti-human AP3M1 (Abcam), polyclonal rabbit anti-human AP3B1 (Proteintech), and monoclonal mouse anti-human lactoferrin (Abcam). Membranes were incubated with primary antibodies in 2.5% non-fat milk in TBS-T followed by incubation with secondary antibody goat anti-rabbit IgG IRDye 800CW (LI-COR Biosciences) or goat anti-mouse IgG IRDye 680RD. Immunostaining for GAPDH was used as a loading control. Quantification of bound antibodies was performed on an Odyssey Infrared Imaging system (LI-COR Biosciences).

## ROS production

NADPH oxidase activity was measured by assaying the hydrogen peroxide production by PMNs derived from blood or from iPSC-derived neutrophils, from either the HPS2 or the WT iPSC line, in response to various stimuli with the Amplex Red kit (Molecular Probes; Life Technologies), as described before (25, 38). In short, PMNs or cultured neutrophils (1 × 10$^6$/ml) were stimulated in Hepes medium (132 mM NaCl, 20 mM Hepes, 6.0 mM KCl, 1.0 mM MgSO4, 1.0 mM CaCl2, 1.2 mM potassium phosphate, 5.5 mM glucose, and 0.5% [wt/vol] human serum albumin, pH 7.4) with opsonized *E. coli* (0.25 × 10$^9$/ml), zymosan (1 mg/ml; Sigma-Aldrich), serum-treated zymosan (1 mg/ml), or PMA (100 ng/ml; Sigma-Aldrich) in the presence of Amplex Red (0.5 $\mu$M) and horseradish peroxidase (1 U/ml). Fluorescence was measured at 30-s intervals for 30 min with the HTS7000+ plate reader (Tecan). The maximal slope of hydrogen

peroxide release was assessed over a 2-min interval to determine the activity of the NADPH oxidase.

## Degranulation

Cells were incubated in Hepes medium at 2.5 × 10$^6$ in a round-bottom 96-well plate and pre-incubated with PAF (1 $\mu$M; Sigma-Aldrich) or cytochalasin B (CytoB, 5 $\mu$g/ml; Sigma-Aldrich) for 5 min, at 450 RPM at 37°C (Thermomixer Comfort; Eppendorf; 96-wells plate insert, Order No. 5363 007.009). After priming, the cells were stimulated with fMLF (1 $\mu$M; Sigma-Aldrich) for 10-min shaking at 37°C. After stimulation, the cells were put on ice, washed once with Hepes buffer, and subsequently stained with antibodies against neutrophil granule markers: FITC-labeled anti-CD66b (clone CLB-B13.0; PeliCluster) and APC-labeled anti-CD63 (clone MX-49; Santa Cruz). Markers to discriminate the different development stages of the iPSC-derived neutrophils (CD15, CD11b, and CD16) were taken along in the FACS analysis. Data are expressed as gMFI.

## Protease activity

To measure protease activity after degranulation, degradation was measured of fluorescent DQ Green BSA (Life Technologies) (25). Cells were resuspended at 2.5 × 10$^6$ cells/ml in Hepes medium and pre-incubated with DQ-BSA and 1 $\mu$M PAF or 5 $\mu$g/ml cytochalasin B for 5 min at 37°C. Cells were subsequently stimulated with 1 $\mu$M fMLF or 100 ng/ml PMA (all stimuli were purchased from Sigma-Aldrich). An unstimulated condition and total protease content by Triton X-100 lysis (1% wt/vol) were taken as controls. A plate reader (emission 535 nm; Tecan) was used to monitor fluorescence for 2-min intervals during 1 h.

## Live-cell imaging

On day 25 of culture, iPSC-derived progenitors were stained for 15 min at 37°C in PBS with 5 $\mu$M of CellTracker Red (Invitrogen). M-CSF–derived macrophages at day 10 were used, either unstained or stained with DiO cell labeling dye (Invitrogen) 1:4,000 in PBS for 30 min at 37°C. Cells were washed with PBS and subsequently put on Stemline II with 1% Pen/strep, 30 ng/ml G-CSF, and 10% heat-inactivated serum. iPSC-derived progenitors were put 3:1 with the M-CSF macrophages, and imaging was performed within 5 min after co-incubation of macrophages and iPSC-derived progenitors and lasted up to 180 min at 37°C, 5% CO$_2$ using a Zeiss Axiovert widefield microscope, or z-stacks were made with an Zeiss LSM 980 Airyscan 2 for confocal imaging, both at 40x magnification. Images were captured with ZEN software (Zeiss). Widefield images were analyzed and compiled into movies using FIJI (version 1.53). Confocal images were analyzed using Imaris (64-bit, version 9.8.2). Three-dimensional reconstruction of cells was performed by calculating the surface of the cells with a smooth surface filter. Surfaces were calculated for M-CSF macrophages or HPS2 cells based on either DiO or CellTracker Red staining, respectively, with a surface area detail of 0.298 $\mu$m. The filter threshold was manually determined for visualization purposes only.

## Statistics

Statistical analysis was performed with GraphPad Prism (version 8) for Windows (GraphPad Software). For each experiment, separate inductions are indicated in the figure legends (e.g., n = 4). Experiments were evaluated by Tukey's or Dunnett's multiple comparison, multiple paired *t* test, or paired two-tailed *t* test. In case of multiple comparison analysis or multiple paired *t* tests, corrected *P*-values were used for the determination of statistical significance. The results are presented as the mean ± SD. Data were considered significant when $P < 0.05$.

## Supplemental materials

Supplementary data contain a table listing all antibodies and chemicals used for FACS analysis. In addition, supplemental figures show a schematic representation of our iPSC culture. Furthermore, we show the characterization of the iPSCs with z-VAD, as well as our gating strategy for FACS analysis. Moreover, we also show additional phenotyping of iPSCs against eosinophils and macrophages, and we included some data from bone marrow samples to compare against iPSC data, as well as a comparison of PMNs from an HPS2 patient against a healthy donor. Finally, we also included the movies from which we took images for Fig 3.

## Supplementary Information

## Acknowledgements

We are very grateful to the patients and parents, as well as the treating physicians for their cooperation and consent. Also many thanks to Mark Hoogenboezem, Simon Tol, and Erik Mul for their technical support. Finally, we would like to thank Hanke L Matlung, Eszter Varga, and Dirk Roos for critical evaluation of data during work discussions. This work was supported by the Dutch Ministry of Health (PPOC 19-22, PPOC 2089), as well as E-Rare-3 JTC2018SF (LADOMICS).

## Author Contributions

SDS Webbers: conceptualization, data curation, formal analysis, validation, investigation, visualization, methodology, project administration, and writing—original draft, review, and editing.
CEM Aarts: conceptualization, formal analysis, investigation, methodology, and writing—original draft.
B Klein: investigation and methodology.
D Koops: formal analysis, investigation, and methodology.
J Geissler: investigation and methodology.
ATJ Tool: conceptualization, investigation, and methodology.
R van Bruggen: conceptualization, data curation, supervision, and methodology.
E van den Akker: conceptualization, data curation, supervision, and methodology.
TW Kuijpers: conceptualization, data curation, supervision, funding acquisition, validation, visualization, methodology, project administration, and writing—original draft, review, and editing.

## Conflict of Interest Statement

The authors declare that the research was conducted in the absence of any commercial or financial relationships that could be construed as a potential conflict of interest.

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
