## [Reviewer comments · Life Science Alliance]

Life Science Alliance

REDUCED MYELOID COMMITMENT AND INCREASED UPTAKE BY MACROPHAGES OF STEM CELL DERIVED HPS2 NEUTROPHILS

Steven Webbers, Cathelijn Aarts, Bart Klein, Dané Koops, Judy Geissler, Anton Tool, Robin van Bruggen, Emile van den Akker, and Taco Kuijpers

DOI: <https://doi.org/10.26508/lsa.202302263>

Corresponding author(s): Steven Webbers, Sanquin and Taco Kuijpers, Sanquin Research and Landsteiner Laboratory, Amsterdam UMC

Review Timeline:

Submission Date:	2023-07-10
Editorial Decision:	2023-08-17
Revision Received:	2023-12-05
Editorial Decision:	2023-12-18
Revision Received:	2023-12-22
Accepted:	2023-12-27

Transaction Report:

August 17, 2023

Re: Life Science Alliance manuscript #LSA-2023-02263-T

Mr. Steven Daniel Sebastiaan Webbers
Sanquin
Department of Molecular Hematology
Amsterdam
Netherlands

Dear Dr. Webbers,

Thank you for submitting your manuscript entitled "REDUCED MYELOID COMMITMENT AND INCREASED UPTAKE BY MACROPHAGES OF STEM CELL DERIVED HPS2 NEUTROPHILS" to Life Science Alliance. The manuscript was assessed by expert reviewers, whose comments are appended to this letter. We invite you to submit a revised manuscript addressing the Reviewer comments.

Thank you for this interesting contribution to Life Science Alliance. We are looking forward to receiving your revised manuscript.

Sincerely,

B. MANUSCRIPT ORGANIZATION AND FORMATTING:

Reviewer #1 (Comments to the Authors (Required)):

In this manuscript by Webbers et al, the authors investigated the characteristics of neutrophils derived from an HPS2 patient iPSC line, which allowed for in-depth analyses of neutrophil differentiation compared to wild-type cells. The plan was based on the well documented phenotype in HPS2 patients of neutropenia, but studying this in patients is difficult - here a cell line model allowed for detection of either defects in neutrophil differentiation or an alternative mechanism for decreased neutrophil numbers in patients. The authors show that derived neutrophils exhibit multiple features of normal neutrophils, including the expression of classic markers, lobulated nuclei and at least some functional responses, including ROS production and granule protein expression. However, they show remarkably reduced numbers of cells but a lack of overall cell death, suggesting neither differentiation itself nor survival of the derived neutrophils is affected by the HPS2 mutation. Interestingly, the derived cells express increased levels of SIGLEC9, and despite several functional responses, the cells showed decreased release of NE. Most interestingly, the population included increased numbers of macrophages, and either these derived macrophages or added WT macrophages phagocytosed the HPS2 iPSC-derived neutrophils, which could explain the observed neutropenia in patients. Overall, the work is well-described and the results are interesting, but there are some issues that should be addressed as follows. There are also a couple of grammatical errors (some of which are identified below), so a careful review of the text is warranted.

Introduction, third paragraph, first sentence, the phrase "but these models have do not phenocopy...." is grammatically incorrect, suggest simply removing the "have".

INTRODUCTION, third paragraph, third sentence, the authors first introduce the reader to the HPS2 iPSC that they generated, but in this sentence they discuss the results of "IPSC-derived CD34 hematopoietic progenitors" and their capacities to exhibit neutrophil development as compared to WT - but which types of progenitors are they referring to, those that were derived from the HPS2 patient? This is confusing and the context of these derived progenitors needs to be carefully described. In addition, the authors mention how the HPS2 IPSC-derived neutrophils phenocopied the circulating HPS2 neutrophils with altered granules, but this becomes confusing - this information should be combined with their description of the HPS2 IPSC-derived neutrophils in the previous (third) paragraph, which will then allow for a smoother transition to the subject of macrophages phagocytosing the CD15+ myeloid cells.

RESULTS, first sentence, the authors refer to reference 6 for making the HPS2-derived iPSC lines, but this is the wrong reference as this does not involve making iPSCs - are they referring to reference 16? Moreover, the authors might want to expand upon the third sentence in this paragraph to step the reader from the results of their previous study (ref 16) into the current studies, including what led them to analyze AP3B1 expression and therefore the results in Suppl Fig. 1. Finally, why is this result buried in a supplemental figure? As a prominent feature of the mutant cells, this should be included in the primary text. One might also argue that, as such an interesting feature, further studies should be performed, specifically RT-PCR assays to determine if gene expression explains the lack of AP3B1 protein expression.

RESULTS, Figure 1A, are the differences in cell counts statistically different? A statistical analyses should be included. This could also be applied to Fig. 1B (and multiple other figures, some of which are indicated below).

Supplemental Figure 2A and RESULTS text (first paragraph), did the authors examine apoptosis markers in the HPS2-derived progenitors as they exhibited decreased growth? Despite the negative result from using the caspase inhibitor, it would be easy to stain the cells for an apoptosis marker (and this could still be an issue despite the lack of rescue by the inhibitor). One might argue this is important to show given the interesting phenotype shown with increased phagocytosis by macrophages (which provides an explanation of the patient neutropenia).

Figs. 1D and E, some indicators of statistical significance in the differences of CD34+ or CD15+ cells should be indicated on the figure and/or figure legend. Also, some explanation of the statistics should be included in the Methods; were the "N=6-7" indicate that 6-7 separate inductions were performed to yield the numbers? This should be clearly described in the Statistics section.

Figure 2D and text in RESULTS referring to this figure, the authors need to back up their statement that SIGLEC9 expression was "much higher" in the HPS2 cells with statistics (p-values). The next statement regarding apoptosis should be backed up with marker expression analyses, despite again the negative result of the caspase inhibitor.

RESULTS, third section under "HPS iPSC-derived neutrophils show a granule defect...", 4th sentence, the phrase "and together with the degranulation defect" is grammatically incorrect.

RESULTS, page 6, last paragraph, the description of M-CSF-derived macrophages engulfing and therefore digesting differentiated HPS2 cells is interesting, but some of the wording is a bit confusing. For example, in the second to last sentence in the last paragraph, the authors mention "which was much more evident with WT cells", but the previous sentences indicate that only the HPS2 cells were engulfed, not the WT cells, so this should be resolved. Also, there are 2 major questions here as to what the HPS2 cells express that would cause increased phagocytosis by the M-CSF-derived macrophages, but also why there were so many more macrophages observed in the HPS2 population. These are two separate observations that become a bit tangled in the description - perhaps a rewording along with quantitative analyses of the increased numbers of macrophages in the HPS2 population would help resolve this confusion.

DISCUSSION, paragraph 4, the authors should further discuss the two very interesting results observed for the HPS2 iPSC-derived population of neutrophils with regards to 1) the identified increased population of macrophages, and 2) the increased capacities of these derived macrophages or the added WT macrophages that phagocytose the derived neutrophils (Fig. 3). What might cause the increased numbers in macrophages in the derived populations from HPS2 iPSCs? The authors mention SIGLEC9, but how might the increased expression of this inhibitory receptor (details of which should also be included) cause increased phagocytosis? This receptor can induce cell death in mature PMNs, so why do they not observe increased apoptosis (this provides motivation for further testing apoptosis, as just inducing the cells with an inhibitor may be insufficient to detect the process)? With a bit more details of these important results, their impact will be improved. These descriptions would complement their comments about hemophagocytosis in the following paragraph.

For Supplemental Figure 3, where is this referred within the text? Reference to Supp.Fig.3 was not found by this reviewer, so this should be inserted (perhaps with the description of derived neutrophils, page 5, middle paragraph).

Referee Cross-Comments: This reviewer agrees with the other reviewer regarding the macrophage phagocytosis data in Figure 3, specifically that the images could simply be showing overlapping images of cells rather than actual phagocytosis. As this is arguably the most novel result from this study, quantifying or at least more carefully presenting the phagocytosis of HPS2 cells by either macrophages derived from the HPS2 iPSCs or the added WT macrophages is important, and should be easy to perform with phagocytosis assays available. This combined with a quantified analyses of the increased macrophage production from the HPS2 iPSCs will add to the impact of the work.

Reviewer #2 (Comments to the Authors (Required)):

The manuscript "Reduced myeloid commitment and increased uptake by macrophages of stem cell derived HPS2 neutrophils" by Steven Webbers et al matures induced pluripotent stem cells isolated from a patient with Hermansky-Pudlak syndrome type 2. These cells are used to determine their efficacy of induction and the phenotype of induced cells. They noted decreased numbers of cells with a neutrophil phenotype relative to control cultures, but that their differentiated phenotype was largely normal. Instead, they find accumulation of phagocyte internalization with accumulation of cellular debris. They conclude HPS2 neutrophil development was slower than normal and suffer from significant macrophage clearance. Overall, this is a sound and informative experimental approach, with an significant and unanticipated outcome. Primarily specific attention to statistical comparisons would strengthen the conclusions forwarded in this manuscript.

Critique:

Overall, this work contributes new and significant knowledge and highlights an advantageous approach to complex human disease. There are presentation defects that impact the robustness of this report.

1. The figures lack statistical comparison. For example, the panels in Fig. 1 includes error bars, but no comparison. The number of replicate experiments is given for panels A and B; if these are the same for the entire figure, this should be stated. In Fig 2, only D lacks the number of experiments, but all lack statistical analysis.
2. Fig. 1B should not display a continuous line as the x axis is not a continuous function.
3. Sup Fig. 2 to show classical apoptosis in not involved is not informative as it does not show the z-VAD-treated cells in comparison to untreated culture. It shows fewer HPS2 cells than WT, but no control cells. Fig. 1B with untreated cells has fewer cells of both types and cannot be used as a comparison.
4. P5 para 1 "SIGLEC9 crosslinking has been found to induce apoptosis" requires a reference.

5. P5 para4 "This corresponds to the lack of CD63 upregulation from azurophilic granules or CD66b from specific granules of mature neutrophils (Fig.2G). Moreover, CD63 was already strongly increased on HPS2 iPSC-derived neutrophils, and together with the degranulation defect." This panel in fact shows both WT and HSP2 cells were indistinguishable, and that neither phenotype (probably) recapitulated the level displayed by PMN. This is not a demonstration of a selective effect of AP3 PMN formation.

6. P5 para 5 "very large size, round nuclear morphology, lipid droplets and uptake of other cells" is used to identify these cells as macrophages. This conclusion requires some flow cytometry to distinguish macrophages from phagocytes.

7. Fig.3 legend. The image in panel 3C is considered to show uptake of an HPS2 ce3II by a M-CSF macrophage. However, it is more likely an image of a macrophage overlying an HPS2 cell as both cells are intact and only an edge is juxtaposed with the macrophage.

Minor:

1. P3par1. The discussion of PMN in infections might include their obligate role as APC in CD8 maturation (e.g. PMID: PMC7791396)

Reviewer #1 (Comments to the Authors (Required)):

Introduction, third paragraph, first sentence, the phrase "but these models have do not phenocopy..." is grammatically incorrect, suggest simply removing the "have".

Thank you for the careful reading of the manuscript for grammatical errors, we have done a thorough check for other errors as well.

INTRODUCTION, third paragraph, third sentence, the authors first introduce the reader to the HPS2 iPSC that they generated, but in this sentence they discuss the results of "iPSC-derived CD34 hematopoietic progenitors" and their capacities to exhibit neutrophil development as compared to WT - but which types of progenitors are they referring to, those that were derived from the HPS2 patient? This is confusing and the context of these derived progenitors needs to be carefully described. In addition, the authors mention how the HPS2 iPSC-derived neutrophils phenocopied the circulating HPS2 neutrophils with altered granules, but this becomes confusing - this information should be combined with their description of the HPS2 iPSC-derived neutrophils in the previous (third) paragraph, which will then allow for a smoother transition to the subject of macrophages phagocytosing the CD15+ myeloid cells.

We agree that the wording should be improved upon in this section, and have rewritten this part of the paragraph by first describing how the model has been used, followed by stating our findings (page 4, lines 18-33 of the revised manuscript).

RESULTS, first sentence, the authors refer to reference 6 for making the HPS2-derived iPSC lines, but this is the wrong reference as this does not involve making iPSCs - are they referring to reference 16? Moreover, the authors might want to expand upon the third sentence in this paragraph to step the reader from the results of their previous study (ref 16) into the current studies, including what led them to analyze AP3B1 expression and therefore the results in Suppl Fig. 1. Finally, why is this result buried in a supplemental figure? As a prominent feature of the mutant cells, this should be included in the primary text. One might also argue that, as such in interesting feature, further studies should be performed, specifically RT-PCR assays to determine if gene expression explains the lack of AP3B1 protein expression.

Our apologies for the incorrect numbering of this reference, which was corrected accordingly in the revised version of the manuscript. We also added the panel with the AP3B1 Western blot in Supplemental Figure 1A to Figure 1 of the main text, as suggested by the reviewer. Although we may agree this feature is indeed highly relevant and hence moved to Figure 1 of the main, we believe additional analysis of the AP3B1 complex subunits is beyond the scope of this paper, as this has been shown by Western blotting (Karampini et al, Haematologica 2019; PMID 30630984) and MassSpec (de Boer et al, Hum Mutat 2017; PMID: 28585318). The immunoblot only serves to show that our HPS2 iPSC line indeed lacks AP3B1 and as a result of the other components from the AP3 complex including AP3M. These papers have been referred to (page 5, lines 5-7 of the revised manuscript).

RESULTS, Figure 1A, are the differences in cell counts statistically different? A statistical analyses should be included. This could also be applied to Fig. 1B (and multiple other figures, some of which are indicated below).

As requested by the reviewer, we have performed statistical tests for all panels and figures where applicable. Details can be found in the figure legends. In the Methods section we reported which tests were performed, what corrections were used for multiple testing, etc. (page 14, lines 18-22 of the revised manuscript)

Supplemental Figure 2A and RESULTS text (first paragraph), did the authors examine apoptosis markers in the HPS2-derived progenitors as they exhibited decreased growth? Despite the negative result from using the caspase inhibitor, it would be easy to stain the cells for an apoptosis marker (and this could still be an issue despite the lack of rescue by the inhibitor). One might argue this is important to show given the interesting phenotype shown with increased phagocytosis by macrophages (which provides an explanation of the patient neutropenia).

In Figure 1D, which shows the percentage of live cells, we based the viability on the absence of AnnexinV staining for PS exposure and low nuclear staining by Hoechst compared to dead cells. This was not explicitly stated in the results section of the manuscript. We have provided an additional statement in the revised manuscript (page 5, line 18).

Figs. 1D and E, some indicators of statistical significance in the differences of CD34+ or CD15+ cells should be indicated on the figure and/or figure legend. Also, some explanation of the statistics should be included in the Methods; were the "N=6-7" indicate that 6-7 separate inductions were performed to yield the numbers? This should be clearly described in the Statistics section.

As requested, a statement is now provided in the Methods section regarding the meaning of "n=6-7" or other numeric examples in the figure legends (Page 14, under the header of "Statistics").

Figure 2D and text in RESULTS referring to this figure, the authors need to back up their statement that SIGLEC9 expression was "much higher" in the HPS2 cells with statistics (p-values). The next statement regarding apoptosis should be backed up with marker expression analyses, despite again the negative result of the caspase inhibitor.

We have added the p-values as requested. Also, the marker expression analyses have been explained and are shown as the combined negative staining for AnnexinV and Hoechst in Figure 1D, just as requested (page 6, line 2-4).

RESULTS, third section under "HPS iPSC-derived neutrophils show a granule defect...", 4th sentence, the phrase "and together with the degranulation defect" is grammatically incorrect.

Once again, thank you for pointing to the incorrect grammar in the manuscript. We have changed the paragraph to accommodate this error (page 6, under the header "HPS2 iPSC-derived neutrophils show a granule defect similar to circulating HPS2 neutrophils")

RESULTS, page 6, last paragraph, the description of M-CSF-derived macrophages engulfing and therefore digesting differentiated HPS2 cells is interesting, but some of the wording is a bit confusing. For example, in the second to last sentence in the last paragraph, the authors mention "which was much more evident with WT cells", but the previous sentences indicate that only the HPS2 cells were engulfed, not the WT cells, so this should be resolved. Also, there are 2 major questions here as to what the HPS2 cells express that would cause increased phagocytosis by the M-CSF-derived

macrophages, but also why there were so many more macrophages observed in the HPS2 population. These are two separate observations that become a bit tangled in the description - perhaps a rewording along with quantitative analyses of the increased numbers of macrophages in the HPS2 population would help resolve this confusion.

Here, the wording is indeed confusing. Events where the HPS2 cells interact and become engulfed were easily observed, where this was not the case for the WT cells. Moreover, we changed the wording accordingly, to clarify that the experiments indeed show that the engulfment by macrophages is intrinsic to the HPS2 cultures (page 7, final paragraph).

DISCUSSION, paragraph 4, the authors should further discuss the two very interesting results observed for the HPS2 iPSC-derived population of neutrophils with regards to 1) the identified increased population of macrophages, and 2) the increased capacities of these derived macrophages or the added WT macrophages that phagocytose the derived neutrophils (Fig. 3). What might cause the increased numbers in macrophages in the derived populations from HPS2 iPSCs? The authors mention SIGLEC9, but how might the increased expression of this inhibitory receptor (details of which should also be included) cause increased phagocytosis? This receptor can induce cell death in mature PMNs, so why do they not observe increased apoptosis (this provides motivation for further testing apoptosis, as just inducing the cells with an inhibitor may be insufficient to detect the process)? With a bit more details of these important results, their impact will be improved. These descriptions would complement their comments about hemophagocytosis in the following paragraph.

As suggested by the reviewer, we have now added our further thoughts on the meaning of increased SIGLEC9 expression to the Discussion (page 8 line 19 to page 9 line 5) of the revised version of our manuscript.

For Supplemental Figure 3, where is this referred within the text? Reference to Supp.Fig.3 was not found by this reviewer, so this should be inserted (perhaps with the description of derived neutrophils, page 5, middle paragraph).

As requested, we have added a reference for this supplemental figure. (page 5, line 19). This has resulted in renumbering of the supplemental figures, since now the z-VAD data in the supplement is referenced after the gating strategy for the flow cytometry.

Referee Cross-Comments: This reviewer agrees with the other reviewer regarding the macrophage phagocytosis data in Figure 3, specifically that the images could simply be showing overlapping images of cells rather than actual phagocytosis. As this is arguably the most novel result from this study, quantifying or at least more carefully presenting the phagocytosis of HPS2 cells by either macrophages derived from the HPS2 iPSCs or the added WT macrophages is important, and should be easy to perform with phagocytosis assays available. This combined with a quantified analyses of the increased macrophage production from the HPS2 iPSCs will add to the impact of the work.

We agree with the reviewers that from the images in Figure 3C it is difficult to discern whether HPS2 cells are simply overlapping or truly inside the 'M-CSF macrophages' due to the nature of widefield microscopy. For this reason, we performed confocal microscopy experiments of which the images are shown in Figure 3D, which show definitive proof of the uptake of HPS2 cells by M-CSF macrophages. Here we used z-stack confocal imaging together with volume rendering in Imaris to show that, over

time, HPS2 cells engage and are taken up into the M-CSF-macrophages. The Methods section contains a more detailed section on how this was performed (see page 14, paragraph 1). We have also added Supplementary Videos to the revised version of the manuscript (Video 1 to 4) from which the still images for both Figure 3C and 3D were taken, where especially the confocal imaging shows very compelling evidence for the uptake of HPS2 cells by the M-CSF macrophages from healthy control donors.

Reviewer #2 (Comments to the Authors (Required)):

The manuscript "Reduced myeloid commitment and increased uptake by macrophages of stem cell derived HPS2 neutrophils" by Steven Webbers et al matures induced pluripotent stem cells isolated from a patient with Hermansky-Pudlak syndrome type 2. These cells are used to determine their efficacy of induction and the phenotype of induced cells. They noted decreased numbers of cells with a neutrophil phenotype relative to control cultures, but that their differentiated phenotype was largely normal. Instead, they find accumulation of phagocyte internalization with accumulation of cellular debris. They conclude HPS2 neutrophil development was slower than normal and suffer from significant macrophage clearance. Overall, this is a sound and informative experimental approach, with an significant and unanticipated outcome. Primarily specific attention to statistical comparisons would strengthen the conclusions forwarded in this manuscript.

Critique:

Overall, this work contributes new and significant knowledge and highlights an advantageous approach to complex human disease. There are presentation defects that impact the robustness of this report.

1. The figures lack statistical comparison. For example, the panels in Fig. 1 includes error bars, but no comparison. The number of replicate experiments is given for panels A and B; if these are the same for the entire figure, this should be stated. In Fig 2, only D lacks the number of experiments, but all lack statistical analysis.

As requested by the reviewer, we have performed statistical tests for all panels and figures where applicable. Details can be found in the figure legends. In the Methods section we reported which tests were performed, what corrections were used for multiple testing, etc. (page 14, lines 18-22 of the revised manuscript)

2. Fig. 1B should not display a continuous line as the x axis is not a continuous function.

Thank you for bringing this to our attention. We have changed this for Figure 1B as well as Figure 3A, since this figure also does not show a continuous function.

3. Sup Fig. 2 to show classical apoptosis in not involved is not informative as it does not show the z-VAD-treated cells in comparison to untreated culture. It shows fewer HPS2 cells than WT, but no control cells. Fig. 1B with untreated cells has fewer cells of both types and cannot be used as a comparison.

We agree with the reviewer that a proper control is indeed important for Supplemental Figure 3A. We have performed the z-VAD cultures side-by-side with a DMSO-treated control. The figure with these data has been added to these comments only. There are two reasons that we left out these controls in this graph in the original manuscript. The first one is that the data points are so close together that it becomes difficult to distinguish the controls from the z-VAD cultures. The second one is that the trends shown in Figure 1B and Supplemental Figure 3A are very similar.

4. P5 para 1 "SIGLEC9 crosslinking has been found to induce apoptosis" requires a reference.

Thank you for spotting this omission. We have added the reference accordingly (page 6, line 5-6 of the revised manuscript).

5. P5 para4 "This corresponds to the lack of CD63 upregulation from azurophilic granules or CD66b from specific granules of mature neutrophils (Fig.2G). Moreover, CD63 was already strongly increased on HPS2 iPSC-derived neutrophils, and together with the degranulation defect." This panel in fact shows both WT and HSP2 cells were indistinguishable, and that neither phenotype (probably) recapitulated the level displayed by PMN. This is not a demonstration of a selective effect of AP3 PMN formation.

During the statistical analyses of our datasets as was requested by the reviewer, we noticed that one of the iPSC-to-neutrophil cultures a 5-fold higher value was present compared to all other datapoints of the same time of neutrophil induction, which falls outside of the SD when using a boxplot for the data on CD63 expression on CD15+ CD11b+ CD16hi WT-30 neutrophils in the degranulation experiment.

Important to note, the shift in expression after stimulation is comparable within this induction to the other inductions, but the actual values are much higher, which is likely due to technical variation between experiments. This influences the variance. Upon deletion of this datapoint (as tested according to the statistical rules on outliers), the difference in CD63 expression of unstimulated HPS2 mature neutrophils compared to WT-30 neutrophils and circulating neutrophils is statistically significantly. We therefore believe that the AP3 defect in our HPS2 iPSC-derived neutrophils recapitulates the CD63 expression as found on HPS2 patient circulating neutrophils. We apologize for this mistake that we should have spotted earlier when statistical analyses would have been performed prior to submission of the manuscript.

6. P5 para 5 "very large size, round nuclear morphology, lipid droplets and uptake of other cells" is

used to identify these cells as macrophages. This conclusion requires some flow cytometry to distinguish macrophages from phagocytes.

We have tried to perform flow cytometric analysis of these macrophages. Due to their large size, alternative FSC/SSC gating was used compared to the example given in Supplemental figure 2. However, we were unable to perform effective gating into a single cell population for these cells. This is probably due to the engulfment and aggregation of other cells in the culture. Additionally, high background fluorescence made it particularly challenging to perform any reasonable analysis even with background corrections. Based on the morphology and high background fluorescence we concluded that it is very likely these cells are macrophages. Comparing the FSC/SSC plots could be done, but since we cannot accurately determine markers on these cells, it is difficult to separate simple aggregates from actual large macrophages full of other cells. We have added a statement in the Methods section under the header “antibodies and flow cytometry” to clarify this.

7. Fig.3 legend. The image in panel 3C is considered to show uptake of an HPS2 cell by a M-CSF macrophage. However, it is more likely an image of a macrophage overlying an HPS2 cell as both cells are intact and only an edge is juxtaposed with the macrophage.

In the blow-up images from the widefield imaging in Figure 3C, there is indeed a juxtaposed macrophage close to the orange arrow in the figure. However, this orange arrow in this top panel indicates two cells which are either inside or underneath the M-CSF macrophage in the top-left of the image. We have moved the orange arrow away from this juxtaposed macrophage, and added another white arrow to indicate the juxtaposed macrophage in the top-left of the blow-up.

Minor:

1. P3par1. The discussion of PMN in infections might include their obligate role as APC in CD8 maturation (e.g. PMID: PMC7791396)

Thank you for the suggestion. We believe the mechanism which is at play in the HPS2 setting to be different from the mechanism described in the paper suggested, and have refrained from mentioning it but added another reference on hemophagocytosis by macrophages instead (REF 30 in the revised manuscript).

December 18, 2023

RE: Life Science Alliance Manuscript #LSA-2023-02263-TR

Mr. Steven Daniel Sebastiaan Webbers
Sanquin
Department of Molecular Hematology
Plesmanlaan 125
Amsterdam 1066CX
Netherlands

Dear Dr. Webbers,

Thank you for submitting your revised manuscript entitled "REDUCED MYELOID COMMITMENT AND INCREASED UPTAKE BY MACROPHAGES OF STEM CELL DERIVED HPS2 NEUTROPHILS". We would be happy to publish your paper in Life Science Alliance pending final revisions necessary to meet our formatting guidelines.

- please make sure the author order in your manuscript and our system match
- please be sure that all authors are listed in the Author Contribution section in the manuscript text
- please consult our manuscript preparation guidelines <https://www.life-science-alliance.org/manuscript-prep> and make sure your manuscript sections are in the correct order
- please note that titles in the system and on the manuscript file must match
- please mark the secondary Corresponding Author as such on the manuscript title page
- please add callouts for Figures S2A-C; S3C; S4B-C; S5A-E to your main manuscript text
- there is a call-out for Figure S1B, and this figure doesn't have panels -- please correct

Figure Checks:

- please add scale bars for Figures 2A and S2B

A. FINAL FILES:

B. MANUSCRIPT ORGANIZATION AND FORMATTING:

Sincerely,

Reviewer #1 (Comments to the Authors (Required)):

1. This manuscript describes results from studies of neutrophils derived from an iPSC line generated from a patient with HPS2, which allowed for identifying a mechanism that causes neutropenia in these patients. The authors convincingly show that the HPS2 iPSCs were severely deficient in neutrophil production but exhibited increased macrophage differentiation, and those neutrophils produced were more susceptible to phagocytosis by macrophages, which is a surprising and novel result.
2. The data strongly support a) the decreased capacity of the HPS2 patient iPSCs to produce neutrophils (but that those produced exhibit multiple neutrophil characteristics), and b) phagocytosis of produced neutrophils, which helps to explain the observed neutropenia in HPS2 patients.
3. The authors have adequately edited the manuscript to address the concerns of this reviewer, and the statistical analyses and presentation in the figures are appreciated.

Reviewer #2 (Comments to the Authors (Required)):

The revised manuscript "Reduced Myeloid Commitment and Increased Uptake by Macrophages of Stem Cell-derived HPS2 Neutrophils" by Dr. Kuijpers et al reports used induced pluripotent stem cells derived from a HPS2 patient to study granulopoiesis. They observed that development into CD15^{POS} cells was reduced, and was associated with cells differentiated into segmented CD11b⁺CD16^{hi} neutrophils. These HPS2 neutrophils mirrored their circulating counterparts and demonstrated increased CD63 expression with impaired degranulation. The decrease in neutrophil yield primarily occurred during the final days of HPS2 iPSC cultures and correlated with CD15^{NEG} macrophage phagocytosis of neutrophils. The conclusion is that HPS2 neutrophil development is affected by a slower rate of development and by macrophage-mediated clearance during neutrophil maturation.

The authors have been responsive and have appropriately addressed each issue raised in the original manuscript. The two primary concerns were a lack of statistical comparisons to support the author's conclusions and insufficient analysis to show PMN-like cells were internalized by phagocytic cells to reduce neutrophil output.

The new submission now includes appropriate analysis to show differences between experimental points. They have included private data to show the comparison between z-VAD-treated cells, with a valid rationale for this choice.

The discussion of the original Panel 2G, with data complication from a single, significantly distinct experiment is reasonable.

A key inference from this work is that PMN abundance can be reduced during maturation by excessive development of functional phagocytes that engulf differentiating cells. New imaging now fully supports this deduction. The new images also suggest the large phagocytes would be particularly difficult to assess by flow cytometry, as I had suggested, but the new images do clearly show the action of phagocytes. While these might be among the subclasses of macrophage phenotypes, the deduction that they are macrophages is reasonable.

December 27, 2023

RE: Life Science Alliance Manuscript #LSA-2023-02263-TRR

Mr. Steven Daniel Sebastiaan Webbers
Sanquin
Department of Molecular Hematology
Plesmanlaan 125
Amsterdam 1066CX
Netherlands

Dear Dr. Webbers,

Thank you for submitting your Research Article entitled "REDUCED MYELOID COMMITMENT AND INCREASED UPTAKE BY MACROPHAGES OF STEM CELL DERIVED HPS2 NEUTROPHILS". It is a pleasure to let you know that your manuscript is now accepted for publication in Life Science Alliance. Congratulations on this interesting work.

DISTRIBUTION OF MATERIALS:

Again, congratulations on a very nice paper. I hope you found the review process to be constructive and are pleased with how the manuscript was handled editorially. We look forward to future exciting submissions from your lab.

Sincerely,
